# Behavioral Nudges to Encourage Appropriate Antimicrobial Use Among Health Professionals in Uganda

**DOI:** 10.3390/antibiotics13111016

**Published:** 2024-10-29

**Authors:** Allison Ross, Philip J. Meacham, J. P. Waswa, Mohan P. Joshi, Tamara Hafner, Sarah Godby, Courtney Johnson, Shilpa Londhe, Dorothy Aibo, Grace Kwikiriza, Hassan Kasujja, Reuben Kiggundu, Michelle Cho, Sarah Kovar, Freddy Eric Kitutu

**Affiliations:** 1Deloitte Consulting LLP, Arlington, VA 22209, USA; allross@deloitte.com (A.R.); pmeacham@deloitte.com (P.J.M.); sgodby@deloitte.com (S.G.); courjohnson@deloitte.com (C.J.); shilpa.londhe@nyulangone.org (S.L.); miccho@deloitte.com (M.C.); sakovar@deloitte.com (S.K.); 2USAID Medicines, Technologies and Pharmaceutical Services (MTaPS) Program, Management Sciences for Health (MSH), Kampala 920102, Uganda; jpwaswa@msh.org (J.P.W.); dorothy.emecu@gmail.com (D.A.); kwikirizagracie8@gmail.com (G.K.); hkasujja@msh.org (H.K.) kajjakiggundu@gmail.com (R.K.); 3USAID Medicines, Technologies and Pharmaceutical Services (MTaPS) Program, Management Sciences for Health (MSH), Arlington, VA 22203, USA; thafner@msh.org; 4Department of Pharmacy, School of Health Sciences, Makerere University, Kampala 920102, Uganda; kitutufred@gmail.com; 5Antimicrobial Stewardship, Optimal Access and Use (ASO) Technical Working Committee, National One Health Platform, Kampala 920102, Uganda

**Keywords:** antimicrobial stewardship, behavior change, antimicrobial prescribing compliance, antimicrobial resistance, AWaRe, behavioral nudges, implementation research, point prevalence survey, clinical guidelines, Uganda

## Abstract

**Background/Objectives:** Antimicrobial resistance (AMR) is a global public health concern exacerbated by inappropriate antimicrobial prescribing practices, particularly in low-resource settings such as Uganda. The research aimed to develop a culturally sensitive behavioral intervention, leveraging a “nudge” strategy, to improve healthcare provider adherence to the 2016 Uganda Clinical Guidelines (UCG 2016) in five Ugandan hospitals. This intervention formed part of broader antimicrobial stewardship initiatives led by the United States Agency for International Development Medicines, Technologies, and Pharmaceutical Services Program. **Methods:** This study employed a mixed-methods approach, combining formative research and behavioral intervention. Guided by the Deloitte Behavioral Insights Framework, the research team conducted key informant interviews to identify prescribing barriers and motivators and developed three suitable behavioral interventions: perceived monitoring, ward leaderboards, and educational workshops. The study evaluated the interventions’ impact through point prevalence surveys (PPS), using the World Health Organization PPS methodology at three stages: pre-intervention, immediate post-intervention, and one-month post-intervention. **Results:** Key behavioral themes across individual, social, environmental, and organizational elements informed the intervention design and implementation. The behavioral intervention package increased antimicrobial prescription compliance with the UCG 2016 from 27% at baseline to 50% immediately post-intervention, though these effects diminished at one-month post-intervention. **Conclusions:** Our study addresses an existing gap in behavioral nudges-based operational research on antimicrobial prescribing in low- and middle-income countries. These results showed an immediate improvement in adherence to the UCG 2016 among healthcare providers in Ugandan hospitals, though the effect was attenuated at one-month follow-up. Despite the attenuation, behavior change presents a feasible, cost-effective, and sustainable approach to improving antimicrobial prescribing practices and addressing AMR.

## 1. Introduction

Antimicrobial resistance (AMR) is one of the top 10 global threats to public health, associated with nearly five million deaths annually [1,2]. Many factors contribute to AMR [3,4,5], demanding a complex, comprehensive, multidisciplinary response.

One contributor to AMR is inappropriate antimicrobial prescribing by medical professionals within health facility settings [1], including primary care centers and hospitals [6]. In these settings, antimicrobial prescribing is inappropriate when it is not in compliance with relevant clinical guidelines [7], such as when antimicrobials are prescribed in the absence of clinical indications [8]. Inappropriate antimicrobial prescriptions often include antimicrobials within the Watch or Reserve categories of the World Health Organization (WHO) Access, Watch, and Reserve (AWaRe) classification, increasing AMR risks [9]. Correcting antimicrobial prescribing behavior is complex [10] but is a crucial and modifiable factor in reducing antimicrobial overuse and misuse, thereby lessening AMR [11]. Knowledge of clinical guidelines by itself appears to have little effect [12,13], but behavioral interventions have shown promise in reducing clinically unnecessary antimicrobial prescribing among healthcare providers in select populations [14,15,16,17]. Most behavioral change studies focus on antimicrobial prescribing in high-income countries [18], using complex interventions such as electronic decision support that may not be replicable in low-resource settings [19]; however, initial studies demonstrate the efficacy of behavioral interventions in low- and middle-income countries as well [19].

To implement an effective behavioral intervention, it is necessary to understand the prevalence of antimicrobial prescriptions, especially in locations where data and surveillance are lacking. The WHO developed a standardized point prevalence survey (PPS) tool to improve antimicrobial use surveillance in resource-constrained settings [20]. In Uganda, AMR is of growing concern, in part because a high proportion of antimicrobial prescriptions may be unnecessary. The Uganda Ministry of Health published the 2016 Uganda Clinical Guidelines (UCG 2016) to inform antimicrobial prescribing [21]. Despite these guidelines, multiple studies have identified both misuse and overuse of antimicrobials in Ugandan clinical settings [22,23]. Previously published research in Ugandan hospitals reported that upwards of 70% of all antimicrobial prescriptions were not compliant with the UCG 2016 [24] and also not in line with reported use among other African countries [25,26]. Given the global concern of AMR and the particularly high prevalence of inappropriate antimicrobial prescribing in Ugandan hospitals, improving provider compliance with the UCG 2016 should be a priority.

Uganda implemented the Antimicrobial Resistance National Action Plan (2018–2023) to enhance antimicrobial stewardship (AMS) and promote the optimal use of antimicrobials [27]. This study builds on ongoing AMS efforts supported by the United States Agency for International Development (USAID) Medicines, Technologies, and Pharmaceutical Services (MTaPS) Program [28,29] to implement a behavior change study to improve antimicrobial prescribing compliance with the UCG 2016 in Ugandan hospitals.

Using a mixed-methods approach across a formative research phase and a behavioral intervention phase, this paper presents a novel social and behavioral “nudge” intervention developed and implemented at five hospitals to encourage compliance with antimicrobial prescribing guidelines among a subset of hospital-based providers in Uganda. The objective of the formative research was to gain an improved understanding of the cultural context around antimicrobial prescribing in Ugandan hospitals, informing the development of a culturally appropriate behavioral intervention. The objective of the behavioral intervention was to increase antimicrobial prescribing compliance with the UCG 2016 among providers in the five Ugandan hospitals.

## 2. Results

The results are presented in two parts—the first part describes the results of the formative research phase (Section 2.1), and the second part describes the results of the behavioral intervention implementation phase (Section 2.2).

### 2.1. Formative Research Phase Results

#### 2.1.1. Behavioral Intervention Elements

Table 1 summarizes the key themes from the key informant interviews organized by Deloitte Behavioral Insights (BI) Framework element and behavioral principles.

#### 2.1.2. Individual Factors

Human decision-making is based on both deliberate and automatic modes of information processing. Individuals often favor automatic processing to make decisions when overwhelmed with information, which can lead to sub-optimal outcomes [30]. Key informants noted that available antimicrobial prescribing guidelines, such as the UCG 2016, are overly complex and difficult to reference quickly. Providers are unlikely to refer to them when diagnosing and prescribing antimicrobials. As key informants explained, hospitals often have “bulky” printed copies of the guidelines rather than electronic versions, making it challenging to reference them in situations that require quick diagnoses. They also described negative perceptions of the UCG 2016, including providers’ view that the guidelines are “outdated” and missing health conditions. Therefore, a behavioral intervention that provides digestible information from the UCG 2016 in an easy-to-use format could influence provider decision-making and encourage providers to reference the guidelines during antimicrobial prescribing processes.

#### 2.1.3. Social Factors

Most individuals make efforts to conform to social norms and expectations, so collective norms can influence the behaviors of those individuals [31]. Key informants suggested that their peers prescribe antimicrobials frequently without adhering to recommendations or guidelines. Even as key informants described themselves as conscientious with antimicrobials, their colleagues have “no problem prescribing antibiotics”, creating a social norm of frequent antimicrobial prescribing. In addition, key informants discussed how social pressures from patients often impact prescribing decisions. Patients generally think that they need medication to get better and feel they “must go home with antibiotics”. As a result, providers who are “not quick to prescribe antibiotics are not so popular” and lose patients as “customers” to other providers. Key informants believed that many patients go to private retail drug shops for prescriptions if they do not receive one at the hospital. While these drug shops are registered and regulated by the Uganda National Drug Authority [32], key informants felt that drug shop chemists are more likely to both overprescribe and underprescribe antimicrobials, potentially worsening patient outcomes and contributing to AMR. Patient behavior is beyond the scope of this study. However, a behavioral intervention that shifts a hospital’s injunctive social norm to one in which AMR is a universal concern and all prescribers are expected to do their part could increase guideline compliance.

#### 2.1.4. Environmental Factors

Because most information processing is automatic (instinctive and unconscious), human behavior is largely shaped by contextual factors and environmental cues [30]. Key informants detailed hospital prescribing environments in which diagnostic tools are time-consuming and expensive, posing a “major hindrance” to appropriate antimicrobial prescribing. Key informants noted that hospitals have “limited diagnostic capacity,” so it can take multiple days to obtain results from samples and blood cultures. Additionally, lab tests are expensive and unaffordable for most patients in Uganda. As a result, providers rarely use lab test results to establish whether a patient has a bacterial infection. Instead, they default to diagnosing and prescribing based on clinical presentation and examination alone. Key informants noted that this creates a tendency to overprescribe antimicrobials, as providers worry about failing to treat a potential bacterial infection. While diagnostic availability is beyond the scope of this study, process changes that slow antimicrobial prescribing could encourage more appropriate prescriptions. Environmental changes, such as educational posters or literature, could also encourage reflection at the time of diagnosis and ensure that AMS is top-of-mind for providers.

#### 2.1.5. Organizational Factors

Individual behaviors within organizations are complex [33]. This is especially true within health systems, as behaviors are often guided by external mandates and policies in addition to individual, environmental, and social factors. Within the study hospitals, key informants noted that hospital leaders express concern with AMR but that this concern rarely translates into organizational policies and programming. Key informants described a variety of existing institutional efforts to combat inappropriate antimicrobial prescribing, including drug prescription surveys, sensitization drives, and continuing medical education (CME) sessions. Hospital Medicines and Therapeutics Committees (MTCs) drive AMS efforts, though key informants noted that these efforts were not consistent. Key informants also remarked that there is not sufficient oversight around antimicrobial prescribing, with relevant training occurring through informal apprenticeships rather than robust training programs. A behavioral intervention that engages hospital and ward leaders could strengthen the overall commitment to AMS. Additionally, a more robust, ongoing AMR education program could influence providers’ prescribing behavior.

### 2.2. Behavioral Intervention Implementation Phase Results

#### 2.2.1. Participant Characteristics

Across the five hospitals, PPS data were collected for 738 unique patients at baseline, accounting for 879 antimicrobial prescriptions (Table 2). After the intervention, data were collected for another 738 unique patients immediately post-intervention and an additional 702 unique patients for longer-term follow-up. No meaningful differences were observed between the two baseline populations other than a small hospital distribution variation (Mengo 32% versus 25%; Rugarama 10% versus 15%). Populations pre- and post-intervention were similarly comprised of three females to two males and a median age of mid-20s, with the population post-intervention having slightly more females (*p* = 0.01). Additionally, the total post-intervention population saw significantly fewer patients from Mengo (*p* < 0.001) and more patients from both St. Francis (*p* < 0.001) and St. Joseph’s Hospitals (*p* = 0.04) compared to pre-intervention, as well as more patients presenting with malaria (*p* = 0.02).

#### 2.2.2. Antimicrobial Prescription Prevalence

Prior to the intervention, PPS data collected across the five hospitals showed that 36 unique antimicrobials were prescribed for a total of 879 prescriptions to 550 patients (out of the 738 total patients at baseline, with an average of 1.2 prescriptions per patient). Of all antimicrobial prescriptions observed at PPS time points, the 15 most prescribed prior to the intervention are visualized in Figure 1 as the proportion of antimicrobial-specific prescriptions out of the total amount of prescriptions pre-intervention compared to the corresponding proportion post-intervention. Ceftriaxone was the most prescribed antimicrobial, accounting for 32% of all pre-intervention prescriptions as well as 32% of all post-intervention prescriptions. There were no significant differences pre- and post-intervention in the prescription proportions of specific antimicrobials, with only a slight increase in metronidazole (+3%), ampicillin (+2%), and amoxicillin (+2%), and a negligible decrease in levofloxacin (−2%) following the intervention. There was no change in the approximate average number of antimicrobials prescribed per patient observed between pre- and post-intervention. Following the intervention, five fewer unique antimicrobials (*n* = 31) were prescribed a total of 1575 times to 1033 patients (out of 1440 patients observed, with an average of 1.1 prescriptions per patient) following the intervention (27 antimicrobials with 746 prescriptions observed immediately post-intervention, and 30 antimicrobials with 829 prescriptions seen at the one-month follow-up).

For the same 15 most prescribed antimicrobials pre-intervention, Table 3 shows the total count of prescriptions observed pre- and post-intervention, along with the proportion of those that adhered to the guidelines for the corresponding diagnosis or indication. Across all hospitals, there was an observed increase in the proportion of prescriptions adhering to the UCG 2016 for most of the antimicrobials, with the greatest impact observed for cloxacillin (55% to 96%), ceftriaxone (23% to 47%), and metronidazole (24% to 32%).

#### 2.2.3. WHO AWaRe Classification

An increase in adherence to the UCG 2016 was observed for many prescribed antimicrobials following the intervention. These antimicrobials are classified as either Access, Watch, Reserve, or Not Recommended, according to the 2023 WHO AWaRe classification of antibiotics [9]. With each observed antimicrobial prescription from the PPS data pre- and post-intervention assigned to one of the AWaRe categories, Figure 2 visualizes the proportions of prescribed antimicrobials in these categories out of the total amount of observed prescriptions pre-intervention compared to the corresponding proportion post-intervention. WHO Access category antimicrobials were prescribed more frequently than those in other categories, accounting for 48% of the total antimicrobials prescribed across all five hospitals pre-intervention and 53% of the total prescribed post-intervention. When evaluating within hospitals, pre- and post-intervention (Figure 2), the WHO Access category antimicrobials remain the most prescribed, except for WHO Watch category antimicrobials accounting for the most prescriptions at Mbale RRH pre-intervention (61%, 186/307), Mengo post-intervention (51%, 186/362), and St. Francis post-intervention (50%, 64/129). Mengo was the only hospital to report the use of WHO Reserve category antimicrobials, accounting for 2% (6/287) of all prescriptions observed pre-intervention and <1% (2/362) of all prescriptions observed post-intervention. WHO Not Recommended antimicrobial combination prescriptions were reported at all five hospitals and antimicrobial combinations in this category increased at every hospital following the intervention. The three combinations reported were cefoperazone/sulbactam (Fytobact^®^), prescribed four times pre-intervention (0.5%) and 60 times post-intervention (4%); flucloxacillin/amoxicillin (Flucamox^®^), prescribed 18 times pre-intervention (2%) and 14 times post-intervention (0.9%); and ampicillin/cloxacillin (Ampiclox^®^), prescribed 17 times pre-intervention (2%) and 23 times post-intervention (2%).

#### 2.2.4. Evaluating Intervention Effects

The intervention’s intended consequence was to improve provider adherence to the UCG 2016 when prescribing antimicrobials for given diagnoses. Without differentiating by specific antimicrobial, Table 4 shows the prescription prevalence and adherence to the UCG 2016 pre- and post-intervention within each of the five hospitals. All five hospitals saw an increase in overall antimicrobial prescribing compliance across post-intervention time periods compared to baseline, with post-intervention compliance 1.5 times the compliance proportion at baseline (from 26.6% to 40.1% post-intervention). Greater increases were observed for the four private, not-for-profit hospitals than for Mbale RRH, which is a public hospital.

Figure 3 further evaluates compliance proportions by the two post-intervention time points within each hospital. At every site, there was a significant and impactful increase in compliance immediately post-intervention (27% at baseline across all hospitals compared to 50% immediately post-intervention), but for most of the sites, this impact diminishes to baseline levels when compliance is assessed at one-month follow-up (31% average compliance across all hospitals).

Compared to baseline, the proportion of compliant prescriptions sustained a significant increase in all wards at the latest follow-up time point (Table 5). The greatest increase in compliance was observed in the pediatric ward (42% to 61%; *p* < 0.001). When controlling for hospitals, these ward-specific compliance increases are still observed at one-month follow-up within each of the five hospitals (the medical wards at St. Francis Hospital, Nkokonjeru, and St. Joseph’s Hospital, Kitgum, did observe a slight decrease in compliance at follow-up, but these differences are not statistically significant). Rugarama Hospital, Kabale, observed the largest increases in compliance proportions on average across all wards, with approximately a two-fold increase sustained at follow-up compared to baseline (5% to 27% for the maternal ward, 44% to 56% for the medical ward, 24% to 73% for the pediatric ward, and 21% to 56% for the surgical ward).

## 3. Discussion

This pilot study is one of the first to develop a multifaceted behavioral intervention and evaluate its effectiveness in motivating compliance with antimicrobial prescribing guidelines among Ugandan hospital providers. The team conducted formative research to inform the design and implementation of the behavioral intervention; this approach not only ensured that the intervention was culturally appropriate and relevant to the study hospitals but also provided insight into broader factors shaping antimicrobial prescribing for future study.

Achieving universal health coverage (UHC) requires deliberate efforts to improve the availability of and access to antimicrobials and preserve their effectiveness [34]. In 2020, Uganda began implementing a roadmap to advance toward UHC, creating an opportunity to strengthen AMS in the country [35] through measures such as standardized antimicrobial therapy across populations and mandated restrictions on antimicrobial use. Moving forward, UHC roadmap actions can encourage AMS by addressing key barriers and motivators to AMS identified in the study’s formative phase (see Table 1).

### 3.1. Uganda Clinical Guidelines Compliance

The overall proportion of antimicrobial prescriptions in compliance with the UCG 2016 was 27% prior to the intervention, in line with previous study findings of 30.1% [24]. This proportion significantly increased to 50% when observed immediately post-intervention. Large within-hospital and within-ward increases in guideline compliance were also observed immediately post-intervention. These immediate effects were attenuated at one-month follow-up but still remained above baseline levels.

While there is a dearth of research into the long-term effects of common behavioral interventions, our results are in line with findings on the short- and long-term impacts of behavioral interventions [36,37,38]. Existing studies indicate that individuals often revert to their prior behaviors in the months following an intervention [39,40]. In part, this may be because the behavioral principles required to initiate behavioral change differ from those required to sustain and maintain behavioral change [41]. For example, the novelty and salience of a new intervention may prompt individuals to change their behavior in the short term [42], while tangible costs and benefits may play a larger role in long-term habit formation [36,41]. Feedback- and monitoring-based behavioral nudges, such as our intervention, often produce immediate consequences and influence individual self-belief and identity, factors which generally encourage sustained motivation and durable behavioral change [36,37,38,41]. That said, ward-level feedback and monitoring may not prompt sufficient individual reflection to motivate long-term behavior change, especially as factors unaddressed by our intervention—such as patient pressure, insufficient laboratory capacity, and financial incentives—push providers toward previous prescribing behaviors [19].

Compliance with the UCG 2016 improved across the study hospitals, regardless of ownership (public vs. private). Four out of the five hospitals demonstrated significant improvements in compliance between the baseline and post-intervention assessments, although the degree of improvements varied between hospitals. Similar variations in AMS outcomes have been demonstrated in other settings in Uganda [28]. Although the reasons for variation among hospitals were not within the study’s scope, internal and external factors specific to each hospital may have contributed to the different compliance levels. Significant improvements in compliance with the UCG 2016 were also observed across all four hospital wards, with the most notable progress seen in the maternal ward. Such ward-based improvements in antibiotic use have been under-documented in Ugandan hospitals, though they have been demonstrated in other African countries [43,44]. The notable improvement in the maternal ward suggests enhanced practices in obstetrics, gynecology, and neonatal care, which could contribute to better maternal, newborn, and child health (MNCH) outcomes. This progress is especially significant given the ongoing MNCH challenges in Uganda [45]. These findings highlight the potential to implement ward-based AMS programs and integrate AMS efforts within MNCH programming to reduce AMR and improve patient outcomes.

Our intervention led to a statistically significant improvement in ceftriaxone, metronidazole, and cloxacillin prescription compliance with the UCG 2016. These results are consistent with findings from similar studies in comparable settings in Uganda [28]. The WHO promotes AMS interventions targeting the optimal use of Watch antimicrobials due to their elevated risk for AMR [46]. Because ceftriaxone is categorized as a Watch antimicrobial [9], increases in prescribing compliance are a positive outcome of our intervention. Moreover, the improvement in ceftriaxone prescribing may have positively influenced the use of metronidazole, as these antimicrobials are commonly administered together for surgical antibiotic prophylaxis (SAP) in Uganda [24,28]. This outcome suggests a promising intervention strategy for addressing the prevalent misuse of antimicrobials associated with SAP. Additionally, the use of cloxacillin, 1 of the 20 most frequently consumed antimicrobials in Uganda [47,48], also significantly improved following the intervention. This positive outcome may be attributed to the limited number of indications for cloxacillin in the UCG 2016, making improvements in guideline compliance easier and further underscoring the effectiveness of antimicrobial-specific AMS initiatives. The lack of statistically significant differences for most antimicrobials may be attributed to several factors, including differences in sample sizes and statistical power, variability in prescribing practices, specific clinical indications for each antimicrobial, and antimicrobial availability and accessibility. However, identifying the causal explanations behind these differences was beyond the scope of our study.

Overall, our intervention effectively generated an immediate improvement in antimicrobial prescribing compliance with the UCG 2016, but future interventions should evaluate whether multi-level approaches generate lasting behavioral change.

### 3.2. WHO AWaRe Classification of Antibiotics

Because our study used the standard WHO PPS methodology, antimicrobial prescription prevalence in Uganda can be compared to other countries. Among patients observed prior to the intervention across the five study hospitals, 74.5% of patients (*n* = 550) were prescribed at least one antimicrobial. This is comparable to a previous WHO PPS study in Uganda showing 73.7% [24] and PPS studies in other low- and middle-income countries [49,50,51]. AMS is crucial to rationalizing antimicrobial prescribing and reducing AMR, leading to improved patient outcomes. The WHO AWaRe classifies antibiotics into four stewardship groups—Access, Watch, Reserve, and Not Recommended—establishing a decisional hierarchy based on the importance of optimal uses and potential for AMR. This study found that a high percentage of pre-intervention prescriptions fell into the WHO Watch category (47%); the most prescribed antimicrobial in the Watch category was ceftriaxone (32%). Frequent prescribing of ceftriaxone and other Watch antimicrobials is in line with Ugandan [24], worldwide [52], and sub-Saharan African findings [53,54,55], raising concerns since evidence suggests that high frequencies of ceftriaxone prescriptions are associated with resistance in hospital settings [56]. A study evaluating factors influencing ceftriaxone prescribing indicated that ceftriaxone was often overprescribed for pneumonia and sepsis [57]. This could suggest that providers continue to prefer ceftriaxone as an antimicrobial treatment for specific indications. While improvements in Access category prescription compliance rates are important to address AMR, AMR containment also requires efforts to restrict the use of Watch and Reserve antimicrobials without compromising the quality of care. AMR containment also requires robust efforts and monitoring to stop prescription of the Not Recommended category of antimicrobials. Evidence suggests that, compared to ceftriaxone, the Access category antimicrobial ampicillin maintains comparable clinical outcomes in community-acquired pneumonia but is associated with significantly lower rates of *Clostridioides difficile* infection [58]. Additionally, patients treated with Access antimicrobials such as oxacillin, nafcillin, and cefazolin have shown no difference in readmission rates compared to those treated with ceftriaxone. Therefore, there remains a need to evaluate the frequent prescription of ceftriaxone and other Watch antimicrobials with high resistance potential. Interventions have been shown to be effective at dissuading inappropriate antimicrobial use or supporting appropriate use in other high-income countries [59,60]. Education-based behavior change interventions, often as part of a multi-pronged approach, have also shown a positive effect on antimicrobial use in low-income countries [14] and could be an appropriate means to dissuade Watch antimicrobial prescribing.

### 3.3. Strengths and Limitations

Our study measured compliance with the UCG 2016, the most recent national prescribing guidelines available at the time of study implementation. During formative research, key informants noted that the UCG 2016 has significant gaps and does not always reflect the latest trends in AMR containment. For example, the UCG 2016 does not include robust recommendations for antimicrobial use during surgery, making it an insufficient reference for common prescribing behaviors such as the use of antimicrobials for surgical prophylaxis. Following the intervention, the Ugandan Ministry of Health published the 2023 UCG [61], but our study does not account for any changes between 2016 and 2023 when measuring compliance.

An additional limitation of our study is the relatively small sample size used for comparisons in the pre- and post-intervention data. This is particularly impactful when using the WHO PPS methodology, which relies on robust sample sizes to accurately reflect population parameters. Limited resources of the field team made it difficult to collect enough data within the WHO PPS-recommended three-week window, affecting our ability to detect statistically significant differences if there were effects to observe. To reach the desired sample size, the team collected two sets of data at each hospital for each of the three time points: baseline, immediate, and follow-up. The two data sets were collected approximately three to six weeks apart and were referred to as time A and time B, respectively. To improve power, the team pooled point prevalence data across pre-intervention times A and B after testing for population differences and determining that these groups were not noticeably different in demographics or underlying conditions. Similarly, we pooled immediate post-intervention time A and B data and follow-up post-intervention time A and B data to have a single group to maximize power when making pre-post comparisons. As a consequence, our study is not able to account for confounding due to the time since intervention. We expect our observed results to be more conservative than the true value because the data suggest that compliance increases immediately post-intervention and then attenuates during follow-up. If we compared the baseline to only immediately post-intervention data with sufficient power, we expect the true effect size to be larger than observed with pooled data. While our findings provide initial insights into the effect of behavioral interventions, future studies with larger sample sizes are needed to validate our results and provide more precise estimates.

The differences observed in demographic and clinical characteristics of patients that reached statistical significance over time—such as gender distribution and comorbidities (e.g., malaria)—may have influenced the results and should be considered when assessing the intervention’s outcomes. These patient differences likely stem from complex factors, such as seasonal variations in patient attendance. However, addressing these factors falls outside the scope of this study and necessitates further research to gain a deeper understanding of how they may impact antimicrobial prescriptions.

Our study did not include a control group, which hinders strong causal inferences about the intervention’s impact. Although pre/post comparisons indicate a positive effect on prescribing behavior, external factors occurring during the study period, such as seasonal variations in disease burdens, may have influenced the results. Additionally, our study implemented a package of three interventions concurrently, so we cannot measure the effect of or attribute impact to specific behavioral interventions. Future studies with a control group are recommended to enhance the evidence for the intervention’s effectiveness.

Our study intentionally included public and private hospitals representing each region in Uganda, though our purposeful sampling limits our ability to generalize study findings to other hospitals in Uganda or similar settings. The five study hospitals had prior experience with AMS programs, with some interventions already in place. This may have influenced healthcare providers’ receptiveness to our behavioral intervention package, potentially limiting the generalizability of the findings. Familiarity with AMS principles might have raised interest in the interventions, affecting provider engagement. Future research should involve conducting baseline assessments of existing AMS core elements to identify specific needs and methods to systematically strengthen AMS programs in healthcare facilities.

One strength of our study is the use of formative research to design the behavioral intervention package. Leveraging best practices for health behavioral interventions [62], the team conducted in-depth qualitative interviews with facility- and national-level stakeholders to understand antimicrobial prescribing behaviors in context. The package of interventions directly addressed key informant interview findings, such as the cognitive load of interpreting complex prescribing guidelines. While the formative research lengthened the duration of our study, requiring multiple facility- and national-level approvals, the resulting interview insights allowed the team to develop a salient, culturally appropriate intervention that was feasible and cost-effective to implement. This minimized the logistical burden placed on hospital intervention teams and encouraged continued interest and sustainability. Additionally, the formative research shed light on complex challenges to appropriate antimicrobial prescribing, such as social pressure from patients and drug promoters. While these were beyond the scope of our intervention, as we focused on healthcare provider behavior rather than patient behavior or broader socioeconomic factors that can significantly influence AMS, our formative research findings can inform future research and broader AMS capacity-building efforts. Addressing these broader factors is essential for enhancing the design and implementation of effective AMS programs, especially in resource-limited settings where sustainability poses a challenge. Incorporating these insights in future research will provide a more comprehensive understanding of the multifaceted nature of AMR and help improve the efficacy of AMS interventions.

Another strength of our study is that the three behavioral interventions were designed to be replicable and logistically feasible in low-resource settings. During formative research, key informants provided feedback on factors that would promote sustainability after intervention implementation. Leveraging their input, the team designed low time-burden interventions that built on existing hospital committees and initiatives. That said, our Round 2 key informant interviews did not include pharmacists and MTC chairpersons due to key informant availability. Because Round 2 interviews focused on intervention impact and feasibility, this gap may have negatively impacted the feasibility of the interventions and their replicability across facilities.

The team also trained MTCs and additional interested staff members to implement the interventions and drive change through intervention teams. Even after the completion of this study, the trained MTCs continued to engage in AMS activities such as ward-based prescription monitoring, educational initiatives, and regular review meetings. This demonstrated successful capacity transfer and motivation, marking a significant step toward the institutionalization of AMS in these hospitals. Empowering MTCs and other capacities throughout intervention design and implementation helped facilitate local ownership and AMS capacity building, laying the foundation for continuous quality improvement long after the study’s conclusion.

### 3.4. Areas of Future Research

During the formative research phase, key informants discussed many challenges to appropriate antimicrobial prescribing, including insufficient lab capacity, patient pressure, and financial incentives. These insights are in line with other studies exploring AMR contributors in Uganda and elsewhere [63,64], though there is insufficient literature on interventions targeting the behavior of patients and/or the general public [19]. Due to time and resource constraints, our behavioral interventions could not tackle these broader contributors to inappropriate antimicrobial prescribing. Further research is needed to explore whether coupling provider-focused behavioral interventions with broader capacity-strengthening activities and investments could promote sustained behavioral change.

Our study focused on antimicrobial prescribing in Ugandan hospitals. Because the majority of global antimicrobial prescribing occurs in primary care settings [65,66,67], future research should also focus on changing provider behaviors outside of the hospital setting, exploring interventions that target professionals in primary care facilities, pharmacies, private retail drug shops, and other environments where patients obtain antimicrobials.

## 4. Materials and Methods

The methods are presented in two parts—the first part describes the methods for the initial formative research (Section 4.1), while the second part details the methods for the subsequent behavioral intervention implementation (Section 4.2).

### 4.1. Formative Research Phase Methods

#### 4.1.1. Study Setting

The study took place from December 2022 to August 2023 in Mbale RRH; Mengo Hospital; Rugarama Hospital, Kabale; St. Francis Hospital, Nkokonjeru; and St. Joseph’s Hospital, Kitgum. Each hospital includes medical, surgical, pediatric, and maternal wards, along with other specialties. Additional hospital characteristics are presented in Table 6. These five hospitals (four private and one public) of varying sizes and geographic regions were purposefully selected due to their engagement with MTaPS AMS programs during the study period.

#### 4.1.2. Behavioral Insights Framework

The Deloitte BI Framework describes how behaviorally informed messaging can be implemented to motivate behavior change (Figure 4). Rooted in theories of behavioral economics, behavior change, behavioral design, psychology, and neuroscience, the BI Framework describes four factors that influence how individuals absorb, process, and react to information: individual, social, environmental, and organizational. Each factor in the BI Framework includes underlying concepts that can be used to understand and encourage behavior:Individual: People are faced with more information than they can process, so simplifying complex tasks, processes, and policies can guide behaviors.Social: Recognizing the importance we place on our connection to others, individuals can be motivated to behave better by comparing their behavior to the behavior of others.Environmental: Because behavior is shaped by our surroundings, prompts and cues can guide behavior so that the desired choice is the easy choice.Organizational: Individual decision-making within organizations is influenced by complex policies and rules, managerial controls, and chains of command, structures that can be shaped to encourage specific behaviors.

**Figure 4 antibiotics-13-01016-f004:**
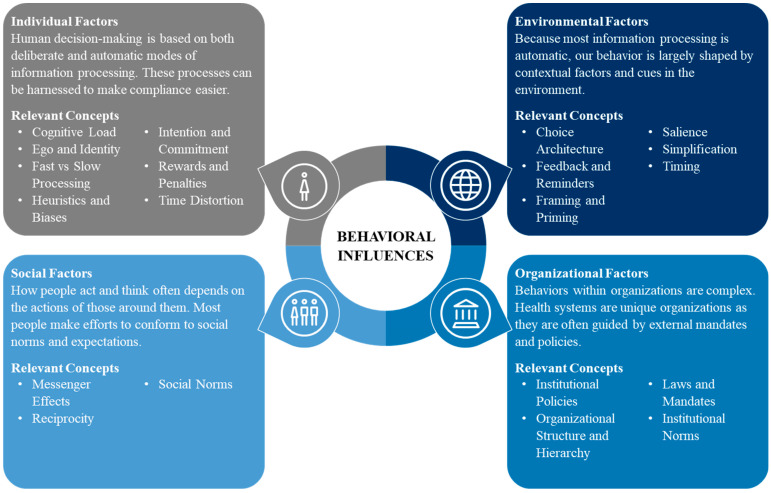
Behavioral Insights Framework.

Developed to incorporate behavioral insights into public policies and government interaction, the framework has been adapted to describe how individuals absorb, process, and react to information in a variety of contexts, such as tax compliance, disaster preparedness, and vaccine adoption [42]. The BI Framework guided our formative research approach to identify interventions that could best influence provider decision-making when prescribing antimicrobials.

#### 4.1.3. Formative Research Design

The research team conducted qualitative, semi-structured key informant interviews to gather information on the individual, social, environmental, and organizational factors that influence antimicrobial prescribing at the five study hospitals, informing the design of a locally appropriate and acceptable behavioral intervention. Key informants were interviewed individually to obtain in-depth, personal perspectives from each participant without the influence of group dynamics. This encouraged participants to speak openly about their experiences and opinions, thereby minimizing the risk of social desirability bias that often arises in group settings.

The team conducted the interviews over two rounds. Round 1 interviews explored BI Framework elements to uncover barriers and motivators that may drive antimicrobial prescribing behavior. The team prepared an interview facilitator guide with questions focused on antimicrobial prescribing processes (e.g., paper versus electronic medical records, internal policies and procedures, training) and the cultural context in which providers prescribe antimicrobials (e.g., patient demand, peer behavior). The team leveraged this information to design behavioral intervention options and presented these options to key informants in Round 2 interviews. Round 2 interview questions focused on the impact and feasibility of two to three potential interventions (e.g., intervention logistics, culturally appropriate language, efficient modes of delivering messaging). Round 1 and Round 2 interview facilitator guides can be found in Appendix A.

The team conducted Rounds 1 and 2 interviews via 30 to 60 min online sessions using the Zoom Video Communications Inc. (Zoom) virtual platform. In two cases, the team conducted interviews in-person because of poor internet connectivity. Each interview included a facilitator who followed the interview facilitator guide to prompt discussion and address all relevant topics. In addition to the facilitator, a note-taker participated in most interviews to capture in-depth insights from the key informants. When the key informant gave permission, the facilitator and note-taker recorded interviews and developed transcripts using Zoom’s built-in transcription feature. All interviews were conducted in English, one of Uganda’s two official languages.

#### 4.1.4. Formative Research Population and Data Collection

For the Round 1 key informant interviews (*n* = 19), the team conducted 17 interviews from 12 December to 20 December 2022 and two additional interviews on 28 March 2023 (Mengo Hospital Round 1 interviews were conducted off-cycle due to approval delays. The research team analyzed and included these results after initial analysis). The team conducted Round 2 interviews (*n* = 7) from 13 March to 28 March 2023.

The target population for the Rounds 1 and 2 key informant interviews included hospital-level stakeholders and national-level stakeholders (Table 7). The hospital stakeholders included officials and healthcare providers who work with antimicrobials or work directly in patient care at each study hospital. The national stakeholders included individuals who support antimicrobial programming in Uganda, such as Ministry of Health staff members and national AMS technical working group members. The team used purposeful sampling, including input from the National AMR Sub-Committee, to enroll key informants in individual interviews. The team recruited potential participants by email and/or phone call. A total of 30 individuals were invited to participate in the interviews. Of these, 19 individuals were interviewed in Round 1, 7 of whom were also interviewed in Round 2 (Table 7).

#### 4.1.5. Formative Research Data Analysis

The team’s behavioral insights experts analyzed the key informant interview notes, audio recordings, and transcripts to identify recurring insights related to antimicrobial prescribing barriers and motivators, as well as significant feedback on the behavioral interventions. The team grouped Round 1 interview insights into behavioral themes based on the BI Framework, describing the individual, social, environmental, and organizational factors that influence providers’ prescribing behavior. The team then reviewed behavioral insights literature to identify evidence-based interventions that could address relevant barriers and motivators identified in the interview themes. The team used Round 2 interview findings to prioritize and further refine the behavioral intervention options to identify those that were feasible, salient, and culturally relevant to the study hospitals.

### 4.2. Behavioral Intervention Implementation Phase Methods

#### 4.2.1. Behavioral Intervention Setting and Design

The team conducted a pilot study with a pre-test/post-test design without a control group. Previous interventions have shown the effectiveness of multifaceted bundled interventions [19], so the team designed a package of three complementary behavioral interventions based on existing literature and formative research findings: perceived monitoring, ward leaderboards, and educational workshops.

The interventions were implemented in the same setting as the formative research phase, with study sites comprising five regional hospitals: Mbale RRH; Mengo Hospital; Rugarama Hospital, Kabale; St. Francis Hospital, Nkokonjeru; and St. Joseph’s Hospital, Kitgum. The interventions targeted all medical professionals in these five hospitals who prescribe antimicrobials to patients (*n* = 200).

The team sought to implement interventions that are logistically feasible and replicable in resource-limited settings. Key informants suggested that close coordination with the hospital MTCs and other clinicians would enhance the sustainability of each intervention, so the team held activation meetings in April and May 2023 to engage MTCs and other hospital staff. The team also established intervention teams at each hospital, consisting of two to five clinicians and staff members who helped coordinate behavioral intervention activities within their hospital on behalf of the MTCs.

The team worked with intervention teams to implement the three interventions over six-week periods at each hospital from July to August 2023. A sample intervention timeline can be found in Appendix A.

Perceived Monitoring: The perceived monitoring intervention communicated to hospital providers that antimicrobial prescribing could be monitored for compliance with the UCG 2016, based on the behavioral insight that the perception of accountability improves compliance with injunctive norms [68]. The team developed a perceived monitoring letter informing providers that the hospital is committed to AMS and that it would monitor prescription compliance with the UCG 2016. The letter also introduced the other interventions as part of the overall monitoring effort. Hospital MTCs shared the letter with all healthcare providers through existing communication channels, including email and/or WhatsApp. The perceived monitoring letter was sent to each hospital in early- to mid-June 2023, commencing the implementation of interventions.

Ward Leaderboards: The ward leaderboard intervention publicly displayed each hospital’s UCG 2016 compliance rate by ward (maternal, medical, pediatric, and surgical), building on the literature showing that audit-and-feedback and peer comparison interventions effectively prompt behavior change [69]. To produce the ward leaderboards, the team trained each hospital’s intervention team to collect abridged WHO PPS survey data on antimicrobial prescribing in the maternal, medical, pediatric, and surgical wards, including diagnoses, antimicrobial prescriptions, and UCG 2016 compliance. The intervention team collected and shared this data with the team weekly. The team conducted quality assurance and analyzed the data every other week, developing ward leaderboards displaying the percentage of compliant prescriptions over two-week periods by ward. The intervention team coordinated with MTCs to print and display these ward leaderboards in each hospital ward. A sample ward leaderboard is shown in Figure 5. Ward leaderboard data collection began in mid-June 2023. Each hospital posted two sets of leaderboards over the course of the intervention in approximately two-week intervals.

Educational Workshops: The educational workshops provided educational content on AMS and AMR, increasing exposure to the UCG 2016. The workshops also served as a forum to publicize ward leaderboard results and share feedback on prescribing compliance efforts. Previous behavioral studies have shown that ongoing education programs can effectively prompt behavior change, especially when bundled with additional interventions [70]. The team coordinated with the intervention team to hold one- to two-hour educational workshops for hospital providers. Where possible, these workshops were scheduled around existing meeting forums, such as MTC plenaries, ward and departmental meetings, CMEs, and clinical rounds, to maximize attendance and engagement. The team provided general content for each session, which focused on the UCG 2016 and ward leaderboard results. The intervention team facilitated in-depth discussions of hospital- and ward-specific strengths and gaps identified in the data. The intervention team also discussed additional antimicrobial-related topics of interest, such as the overuse of antimicrobials for malaria patients. Table 8 shows hospital-specific educational workshop topics. Each hospital received two educational workshops, which began in mid- to late-July and generally followed ward leaderboard publication.

#### 4.2.2. Behavioral Intervention Population and Data Collection

The primary aim of the intervention phase was to observe a significant increase in antimicrobial prescription appropriateness for given diagnoses after implementing the multifaceted behavioral intervention compared to baseline. Appropriateness was defined as prescription adherence to the UCG 2016.

To measure the intervention’s effects, the team used the WHO Methodology for PPS on Antibiotic Use in Hospitals, version 1.1, which “provides a standardized methodology for use in low-, middle- and high-income countries to estimate the prevalence of antibiotic use in hospitals” [20] and is designed with flexibility in mind for resource-limited settings.

In-country team members were trained on the WHO PPS methodology and data privacy prior to data collection at each of the five hospitals. The team collected data on core variables in the PPS tool related to hospital (e.g., type, size), ward (e.g., type, number of patients), patient (e.g., demographic information, underlying clinical variables), antimicrobial (prescription details), and indication (diagnosis variables). In coordination with MTCs, team members collected data from patients who were admitted to the medical, pediatric, surgical, and maternal wards of the five study hospitals before 8:00 am on the study day, as well as patients discharged on the study day. The team excluded data for patient types listed in the WHO PPS protocol, including patients admitted after 8:00 am and patients in long-term care.

The team collected PPS data in two rounds, each at three discrete time points: pre-intervention baseline, immediate post-intervention, and follow-up at one-month post-intervention (Table 9).

#### 4.2.3. Behavioral Intervention Data Analysis

For each time point, time A and time B data were pooled to ensure sufficient statistical power when making pre- and post-intervention comparisons, testing for no significant differences between populations prior to pooling data. Categorical data are summarized as counts and percentages, utilizing Fisher’s exact tests or Pearson Chi-Square tests when evaluating associations. Median values and IQRs were presented for continuous data. Population differences of these data were analyzed using the non-parametric Mann–Whitney–Wilcoxon test because the data were not normally distributed. A two-tailed *p*-value of <0.05 was considered statistically significant. All analyses were performed using R Statistical Software (v4.2.1; R Core Team 2022) or Microsoft Excel (v16.83; Microsoft Office 365).

### 4.3. Study Ethics and Approval

Ethical approval for the study (including conducting interviews, implementing the intervention, and collecting data) was obtained through the Makerere University Institutional Review Board (Registration No. MAKSHSREC-2022-293) and the Uganda National Council for Science and Technology (Registration No. HS2429ES). The Uganda Ministry of Health provided administrative clearance for the study (Reference No. ADM.130/313/05). No patient identifiers were recorded during data collection.

## 5. Conclusions

Our study addresses an existing gap regarding behavioral nudges-based operational research on antimicrobial prescribing in low- and middle-income countries. These results show that this novel behavioral “nudge” intervention package delivers a short-term effect in improving antimicrobial prescribing adherence to the UCG 2016 among providers at large hospitals in Uganda, an effect that may be maintained through continued engagement of local MTCs and AMS bodies. Because these interventions are feasible, low-cost, and replicable in low-resource settings, our study presents a blueprint that hospitals in such settings can adopt or adapt to implement behavioral interventions, scaling potential impact on antimicrobial prescribing behavior and AMR.

This operational research shows promise in achieving Uganda’s Antimicrobial Resistance National Action Plan objective to promote optimal access and use of antimicrobials [27]. Moreover, these results strengthen the evidence base that social and behavior change interventions successfully modify a variety of provider behaviors, leading to improved clinical practices and enhanced patient outcomes.

## Figures and Tables

**Figure 1 antibiotics-13-01016-f001:**
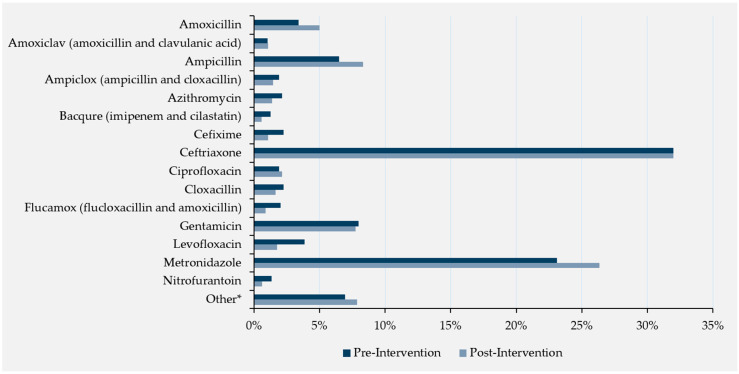
Most prescribed antimicrobials as a proportion of all prescribed antimicrobials, pre- and post-intervention. Notes: * Other consists of the following 21 antimicrobials: linezolid, ceftazidime, pisa (piperacillin and tazobactam), amikacin, erythromycin, meropenem, fytobact (cefoperazone and sulbactam), cefuroxime, cefotaxime, cefazolin, benzathine benzylpenicillin, piperacillin, ofloxacin, doxycycline, phenoxymethylpenicillin, ornidazole, clindamycin, tinidazole, chloramphenicol, moxifloxacin, co-trimoxazole.

**Figure 2 antibiotics-13-01016-f002:**
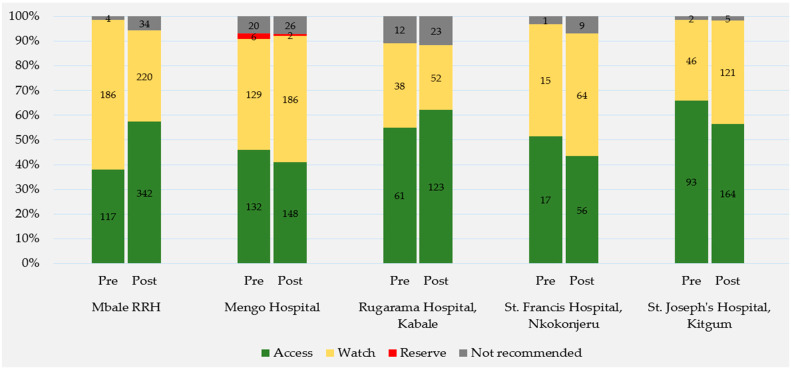
Antimicrobials prescribed per hospital, pre- and post-intervention, by WHO AWaRe classification.

**Figure 3 antibiotics-13-01016-f003:**
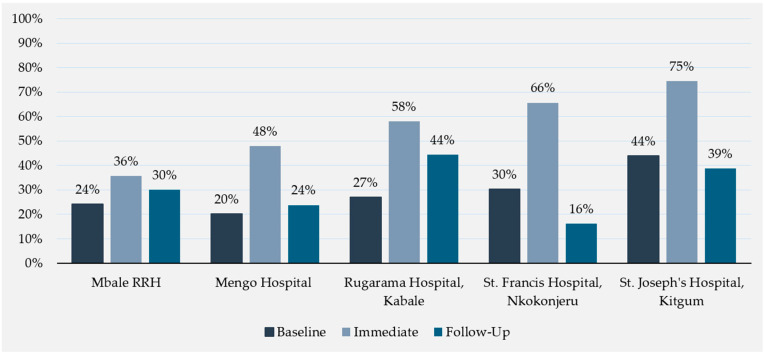
Antimicrobials prescribed in compliance with the UCG 2016 by hospital and time point.

**Figure 5 antibiotics-13-01016-f005:**
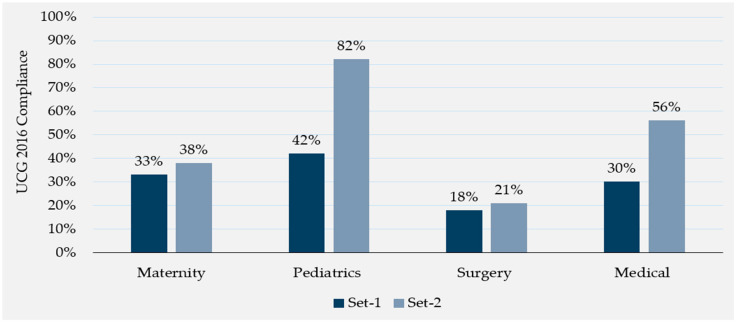
Sample ward leaderboard for St. Joseph Hospital, Kitgum.

**Table 1 antibiotics-13-01016-t001:** Key informant interview findings.

BI Framework Element	Behavioral Themes of Key Informant Interview Feedback
Individual	Time distortion: Providers are faced with the present need to provide care quickly and may prioritize short-term demands over long-term healthcare outcomes. Cognitive load: Providers are often unfamiliar with the UCG 2016 or find the guidelines to be outdated/complex. Heuristics: Access to validating laboratory (lab) results is not always available, so providers must diagnose and prescribe based on clinical presentation alone.
Social	Social norms: The normalized behavior among providers is to prescribe antimicrobials—either due to commercial pressure or because diagnostic capabilities are limited. Social pressure: Patients like antimicrobials and expect to receive a prescription, leading providers to feel pressured to prescribe so they do not lose a patient as a “customer”. Messenger effects: Patients have access to many influential messengers, including drug companies, advertisements, peers, and off-market antimicrobial sellers.
Environmental	Salience: Antimicrobial prescription guidelines are not commonly known or are dismissed as outdated or complex. Choice architecture: There is no standardized protocol for authorizing an antimicrobial prescription (e.g., no lab requirement). Framing and priming: Providers receive direct or indirect pressure from drug promoters and hospital staff to prescribe antimicrobials.
Organizational	Institutional policies: Broader standards established by international health bodies are underrecognized or have little trickle-down to providers. Organizational structure: There is a lack of organizationally provided clinician training and structured, rigorous curriculums around AMS.

**Table 2 antibiotics-13-01016-t002:** Demographic and clinical characteristics of patients by time point.

Variable	Pre-Intervention Time Point A (*n* = 356)	Pre-Intervention Time Point B (*n* = 382)	Total Pre-Intervention (*n* = 738)	Immediate (*n* = 738)	Follow-Up (*n* = 702)	Total Post-Intervention (*n* = 1440)
Demographics						
Female	215 (60%)	204 (53%)	*419 (57%)*	458 (62%)	442 (63%)	*900 (63%) ^+^*
Male	141 (40%)	178 (47%)	*319 (43%)*	280 (38%)	260 (37%)	*540 (38%) ^+^*
Age, median (interquartile range (IQR))	25 (7–43)	27 (10.25–45)	*26 (9–44)*	23 (5–38.75)	28 (12–45)	*25 (8–41)*
Hospital ownership						
Public	143 (40%)	140 (37%)	*283 (38%)*	285 (39%)	253 (36%)	*538 (37%)*
Private not-for-profit	213 (60%)	242 (63%)	*455 (62%)*	453 (61%)	449 (64%)	*902 (63%)*
Hospital						
Mbale Regional Referral Hospital (RRH)	143 (40%)	140 (37%)	*283 (38%)*	285 (39%)	253 (36%)	*538 (37%)*
Mengo Hospital	115 (32%)	97 (25%)	*212 (29%) **	146 (20%)	154 (22%)	*300 (21%) ^+^*
Rugarama Hospital, Kabale	34 (10%)	56 (15%)	*90 (12%) **	92 (12%)	91 (13%)	*183 (13%)*
St. Francis Hospital, Nkokonjeru	7 (2%)	17 (4%)	*24 (3%)*	68 (9%)	47 (7%)	*115 (8%) ^+^*
St. Joseph’s Hospital, Kitgum	57 (16%)	72 (19%)	*129 (17%)*	147 (20%)	157 (22%)	*304 (21%) ^+^*
Ward						
Maternal	97 (27%)	96 (25%)	*193 (26%)*	204 (28%)	199 (28%)	*403 (28%)*
Medical	71 (20%)	97 (25%)	*168 (23%)*	190 (26%)	189 (27%)	*379 (26%)*
Pediatric	110 (31%)	95 (25%)	*205 (28%)*	221 (30%)	153 (22%)	*374 (26%) ^^^*
Surgical	78 (22%)	94 (25%)	*172 (23%)*	123 (17%)	161 (23%)	*284 (20%) ^^^*
Underlying patient condition						
Central catheter	1 (0.3%)	2 (0.5%)	*3 (0.4%)*	4 (0.5%)	4 (0.6%)	*8 (0.6%)*
Peripheral catheter	337 (95%)	358 (94%)	*695 (94%)*	694 (94%)	689 (98%)	*1383 (96%) ^^^*
Urinary catheter	9 (3%)	25 (7%)	*34 (5%) **	34 (5%)	39 (6%)	*73 (5%)*
Intubation	2 (1%)	4 (1%)	*6 (1%)*	8 (1%)	15 (2%)	*23 (2%)*
Malaria	48 (13%)	43 (11%)	*91 (12%)*	153 (21%)	80 (11%)	*233 (16%) ^^+^*
Tuberculosis	5 (1%)	5 (1%)	*10 (1%)*	3 (0%)	6 (1%)	*9 (0.6%)*
Human immunodeficiency virus	7 (2%)	13 (3%)	*20 (3%)*	18 (2%)	28 (4%)	*46 (3%)*
Chronic obstructive pulmonary disease	3 (1%)	4 (1%)	*7 (1%)*	4 (1%)	7 (1%)	*11 (0.8%)*
Malnutrition	14 (4%)	12 (3%)	*26 (4%)*	45 (6%)	17 (2%)	*62 (4%) ^^^*

The numbers in the Total Pre-Intervention and Total Post-Intervention columns are italicized to indicate that they are summaries of the two preceding columns. * Statistically significant difference between time A and time B pre-intervention populations. ^^^ Statistically significant difference between immediate and follow-up post-intervention populations. **^+^** Statistically significant difference between pre- and post-intervention total populations.

**Table 3 antibiotics-13-01016-t003:** Antimicrobials prescribed in compliance with the UCG 2016, pre- and post-intervention.

	Total Prescription at Baseline (*n* = 879)	Baseline Compliance (Time A + B)	Total Prescriptions Post-Intervention (*n* = 1575)	Post-Intervention Compliance (Immediate + Follow-Up)
**Antimicrobial**				
Ceftriaxone	281	64 (23%)	504	238 (47%) ***
Metronidazole	203	48 (24%)	415	131 (32%) *
Gentamicin	70	33 (47%)	122	61 (50%)
Ampicillin	57	27 (47%)	131	64 (49%)
Levofloxacin	34	4 (12%)	28	2 (7%)
Amoxicillin	30	8 (27%)	79	33 (42%)
Cefixime	20	2 (10%)	17	0 (0%)
Cloxacillin	20	11 (55%)	26	25 (96%) ***
Azithromycin	19	4 (21%)	22	5 (23%)
Flucamox (flucloxacillin and amoxicillin)	18	3 (17%)	14	2 (14%)
Ampiclox (ampicillin and cloxacillin)	17	0 (0%)	23	1 (4%)
Ciprofloxacin	17	6 (35%)	34	16 (47%)
Nitrofurantoin	12	11 (92%)	10	9 (90%)
Bacqure (imipenem and cilastatin)	11	0 (0%)	9	0 (0%)
Amoxiclav (amoxicillin and clavulanic acid)	9	4 (44%)	17	6 (35%)
Other ^	61	9 (15%)	124	40 (32%) *

*** Statistically significant at *p* < 0.001. * Statistically significant at *p* < 0.05. ^^^ Other consists of the following 21 antimicrobials: linezolid, ceftazidime, pisa (piperacillin and tazobactam), amikacin, erythromycin, meropenem, fytobact (cefoperazone and sulbactam), cefuroxime, cefotaxime, cefazolin, benzathine benzylpenicillin, piperacillin, ofloxacin, doxycycline, phenoxymethylpenicillin, ornidazole, clindamycin, tinidazole, chloramphenicol, moxifloxacin, co-trimoxazole.

**Table 4 antibiotics-13-01016-t004:** Antimicrobials prescribed in compliance with the UCG 2016, pre- and post-intervention, by hospital.

	Total Prescriptions at Baseline (*n* = 879)	Baseline Compliance (Time A + B)	Total Prescriptions Post-Intervention (*n* = 1575)	Post-Intervention Compliance (Immediate + Follow-Up)
Hospital				
Mbale RRH	307	74 (24%)	596	196 (33%) **
Mengo Hospital	287	58 (20%)	362	128 (35%) ***
Rugarama Hospital, Kabale	111	30 (27%)	198	99 (50%) ***
St. Francis Hospital, Nkokonjeru	33	10 (30%)	129	54 (42%)
St. Joseph’s Hospital, Kitgum	141	62 (44%)	290	156 (54%) *

*** Statistically significant at *p* < 0.001. ** Statistically significant at *p* < 0.01. * Statistically significant at *p* < 0.05.

**Table 5 antibiotics-13-01016-t005:** Antimicrobials prescribed in compliance with the UCG 2016, pre- and post-intervention, by hospital and ward.

	Total Prescriptions at Baseline (*n* = 879)	Baseline Compliance (Time A + B)	Total Prescriptions Post-Intervention (*n* = 1575)	Post-Intervention Compliance (Immediate + Follow-Up)
Ward				
Maternal	297	34 (11%)	577	132 (23%) ***
Medical	165	62 (38%)	299	167 (56%) ***
Pediatric	223	93 (42%)	370	226 (61%) ***
Surgical	194	45 (23%)	329	108 (33%) *
Hospital and ward				
Mbale RRH	307		596	
Maternal	117	16 (14%)	270	53 (20%)
Medical	51	16 (31%)	103	54 (52%) *
Pediatric	57	23 (40%)	118	57 (48%)
Surgical	82	19 (23%)	105	32 (31%)
Mengo Hospital	287		362	
Maternal	114	9 (8%)	111	16 (14%)
Medical	42	5 (12%)	53	27 (51%) ***
Pediatric	77	36 (47%)	111	67 (60%)
Surgical	54	8 (15%)	87	18 (21%)
Rugarama Hospital, Kabale	111		198	
Maternal	19	1 (5%)	63	17 (27%) *
Medical	39	17 (44%)	63	35 (56%)
Pediatric	34	8 (24%)	40	29 (73%) ***
Surgical	19	4 (21%)	32	18 (56%) *
St. Francis Hospital, Nkokonjeru	33		129	
Maternal	19	2 (11%)	26	9 (35%)
Medical	9	6 (67%)	22	10 (46%)
Pediatric	5	2 (40%)	37	23 (62%)
Surgical	0	N/A	44	12 (27%)
St. Joseph’s Hospital, Kitgum	141		290	
Maternal	28	6 (21%)	107	37 (35%)
Medical	24	18 (75%)	58	41 (71%)
Pediatric	50	24 (48%)	64	50 (78%) ***
Surgical	39	14 (36%)	61	28 (46%)

*** Statistically significant at *p* < 0.001. * Statistically significant at *p* < 0.05.

**Table 6 antibiotics-13-01016-t006:** Study site characteristics.

Hospital Name	Region	Size	Public/Private
Mbale RRH	Eastern	450 beds	Public
Mengo Hospital	Central	300 beds	Private
Rugarama Hospital, Kabale	Western	150 beds	Private
St. Francis Hospital, Nkokonjeru	Central	60 beds	Private
St. Joseph’s Hospital, Kitgum	Northern	280 beds	Private

**Table 7 antibiotics-13-01016-t007:** Key informant characteristics.

Interviewee Number	Position Title	Hospital or National Level	Interview Round
1	Epidemiologist	National	Round 1
2	Product Safety Director	National	Round 1
3	Allied Health Professionals Council Member	National	Round 1
4	Clinical Pharmacist and AMR Committee Member	National	Round 1
5	Physician	National	Round 1
6	Department of Microbiology Head	National	Round 1
7	Department of Microbiology Member	National	Round 1
8	Physician	National	Round 1, 2
9	Medical Director	Hospital	Round 1, 2
10	Pharmacist	Hospital	Round 1
11	Medical Director	Hospital	Round 1, 2
12	Pharmacist	Hospital	Round 1
13	Medical Director	Hospital	Round 1, 2
14	MTC Chairperson	Hospital	Round 1
15	Medical Officer	Hospital	Round 1, 2
16	MTC Chairperson	Hospital	Round 1
17	MTC Vice Secretary	Hospital	Round 1, 2
18	Head of Pharmacy	Hospital	Round 1
19	Hospital Director	Hospital	Round 1, 2

**Table 8 antibiotics-13-01016-t008:** Educational workshop topics.

Hospital	Hospital-Specific Educational Workshop Topics
Mbale RRH	Antibiotic use in malaria management Current practice and evidence on fixed-dose combination (FDC) antibiotics Azithromycin and fluoroquinolones in the UCG 2016 SAP and cesarean section Surgical site infection (SSI) prevention Antibiotic use in wound care
Mengo Hospital	FDC antibiotics Azithromycin and cephalosporins in the UCG 2016 SAP and SSI prevention
Rugarama Hospital, Kabale	FDC antibiotics Treatment of sexually transmitted infections and urinary tract infections Clarithromycin and azithromycin in the UCG 2016 SAP and SSI prevention Antibiotic polypharmacy
St. Francis Hospital, Nkokonjeru	FDC antibiotics Best practice for the use of metronidazole and azithromycin Overview of the hospital’s MTC SAP and SSI prevention
St. Joseph’s Hospital, Kitgum	FDC antibiotics Antibiotic use in wound infection Antibiotic use in malaria management Use of piperacillin and tazobactam in surgery Azithromycin and fluoroquinolones in the UCG 2016 SAP and SSI prevention

**Table 9 antibiotics-13-01016-t009:** Data collection timeframes at study sites.

Time Point	Round	Data Collection Dates *	Patients (N)
Pre-intervention baseline	Time A	12–21 December 2022; 25–26 March 2023 for Mengo	356
Pre-intervention baseline	Time B	13–19 February 2023; 10–11 May 2023 for Mengo	382
*Pre-intervention baseline*	*Total*		*738*
Immediate post-intervention	Time A	28 August–8 September 2023	402
Immediate post-intervention	Time B	11–22 September 2023	336
*Immediate post-intervention*	*Total*		*738*
One-month follow-up	Time A	25 September–20 October 2023	339
One-month follow-up	Time B	23 October–8 December 2023	363
*One-month follow-up*	*Total*		*702*

The Pre-intervention baseline, Immediate post-intervention, and One-month follow up rows are italicized to indicate that they are summaries of the two preceding rows. * Mengo Hospital baseline data collection was conducted after other facilities due to additional hospital-specific approval requirements.

## Data Availability

The raw data supporting the conclusions of this article will be made available by the authors upon request.

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
