# Peer review of "Behavioral Nudges to Encourage Appropriate Antimicrobial Use Among Health Professionals in Uganda"

_antibiotics, 2024, doi:10.3390/antibiotics13111016_

Round 1

Reviewer 1 Report

Comments and Suggestions for Authors

1. I would like to express my gratitude for the opportunity to review the manuscript entitled Behavioral Nudges to Encourage Appropriate Antimicrobial Use Among Health Professionals in Uganda” for possible publication in Antibiotics. However, there are some concerns that must be addressed in order to be considered for publication.

2. I noticed that your abstract currently contains 380 words, which exceeds the recommended limit of approximately 250 words. I recommend revising it to ensure it is more concise while still effectively conveying the key findings and significance of your study.

3. I recommend avoiding the division into subsections (1.1, 1.2, 1.3). It would be more effective to present the introduction as a cohesive narrative that highlights the importance and context of your study. 

4. Additionally, the information of study ethics and approval is better suited for the methods section or included in the Institutional Review Board Statement.

5. I suggest incorporating a normality test to ensure that the assumptions for the statistical tests employed are adequately met.

6. Since the median is typically associated with non-parametric data, it would be more appropriate to clarify that you used the Mann-Whitney-Wilcoxon test for continuous data that do not meet normality assumptions, while the Student's t-test should be reserved for normally distributed data.

7. It is essential to validate the questionnaire used in your study, particularly concerning the participants' language backgrounds. Since the official languages of Uganda are English and Swahili, I recommend describing any previously validated questionnaires utilized in similar fields and explaining the rationale behind your chosen methodological approaches.

8. I would like to point out that the demographic and clinical characteristics of patients showed statistically significant differences by time point, yet this important finding has not been adequately discussed in your manuscript.

9. The experimental results concerning the proportion of the most prescribed antimicrobials and the antimicrobials prescribed in compliance with UCG 2016, both pre- and post-intervention, have not been adequately discussed in your manuscript. 

10. Additionally, I would like to highlight that your manuscript primarily focuses on ceftriaxone, while the results for metronidazole and cloxacillin also show statistical significance. It would be beneficial to address why many antimicrobials did not demonstrate statistical significance.

11. also noticed that the experimental results concerning the antimicrobials prescribed in compliance with the UCG 2016, both pre- and post-intervention, categorized by hospital and ward, have not been discussed in your manuscript. Including this analysis would enhance the understanding of the intervention's impact across different settings.

12. I recommend ensuring that the font used in each figure is uniform throughout the manuscript. Consistency in font style and size will improve the overall presentation and readability of your figures, enhancing the clarity of your findings. also suggest enhancing the figure legend for Figure 5 to ensure it reflects a more academic tone.

Author Response

RESPONSES TO REVIEWER 1:
1. I would like to express my gratitude for the opportunity to review the manuscript entitled “Behavioral Nudges to Encourage Appropriate Antimicrobial Use Among Health Professionals in Uganda” for possible publication in Antibiotics. However, there are some concerns that must be addressed in order to be considered for publication.

Authors’ response

We appreciate the time and effort you have put into reviewing our paper. We are thankful for your suggestions and committed to addressing the concerns you have raised and are confident that your feedback has strengthened the manuscript. Please find below our detailed responses on how we have addressed your feedback and made revisions.

2. I noticed that your abstract currently contains 380 words, which exceeds the recommended limit of approximately 250 words. I recommend revising it to ensure it is more concise while still effectively conveying the key findings and significance of your study.

Authors’ response

This has been done (new word count: 254). (lines 21-57).  

3. I recommend avoiding the division into subsections (1.1, 1.2, 1.3). It would be more effective to present the introduction as a cohesive narrative that highlights the importance and context of your study. 

Authors’ response

The subsections have been removed from the introduction section. 

4. Additionally, the information of study ethics and approval is better suited for the methods section or included in the Institutional Review Board Statement.

Authors’ response

This has been moved to the appropriate methods section. (line 790-796)   

5. I suggest incorporating a normality test to ensure that the assumptions for the statistical tests employed are adequately met.

Authors’ response

The only statistical comparison of continuous data is median age in Table 2, which are not normally distributed. No statistical test of normality was performed for the distribution of age; however, histograms and plots showed that age distributions at each time point were right-skewed (median > mean). Section 4.2.3 Data Analysis is updated to show that comparisons based on age were tested using only the appropriate non-parametric test. (line 781-783)

6. Since the median is typically associated with non-parametric data, it would be more appropriate to clarify that you used the Mann-Whitney-Wilcoxon test for continuous data that do not meet normality assumptions, while the Student's t-test should be reserved for normally distributed data.

Authors’ response

This has been clarified accordingly and the changes are reflected in lines 781-786.  

7. It is essential to validate the questionnaire used in your study, particularly concerning the participants' language backgrounds. Since the official languages of Uganda are English and Swahili, I recommend describing any previously validated questionnaires utilized in similar fields and explaining the rationale behind your chosen methodological approaches.

Authors’ response

Thank you for your suggestion. We have realized that our use of the word “questionnaire” may be misleading. We did not administer a questionnaire – we simply created and used a facilitator guide to conduct qualitative interviews to better understand the context of antimicrobial prescribing from a variety of perspectives. We have therefore refined our description of this process to refer to a qualitative facilitator guide and have noted that all interviews were conducted in English. (line 634, 642, 646, and 650-651)

8. I would like to point out that the demographic and clinical characteristics of patients showed statistically significant differences by time point, yet this important finding has not been adequately discussed in your manuscript.

Authors’ response

A paragraph addressing this important issue has been added in the discussion section. (line 495-502)  

9. The experimental results concerning the proportion of the most prescribed antimicrobials, and the antimicrobials prescribed in compliance with UCG 2016, both pre- and post-intervention, have not been adequately discussed in your manuscript. 

Authors’ response

This has been addressed in the discussion section. (line 405-424)  

10. Additionally, I would like to highlight that your manuscript primarily focuses on ceftriaxone, while the results for metronidazole and cloxacillin also show statistical significance. It would be beneficial to address why many antimicrobials did not demonstrate statistical significance.

Authors’ response

This has been addressed in the discussion section. (line 411-424)  

11. I also noticed that the experimental results concerning the antimicrobials prescribed in compliance with the UCG 2016, both pre- and post-intervention, categorized by hospital and ward, have not been discussed in your manuscript. Including this analysis would enhance the understanding of the intervention's impact across different settings.

Authors’ response

This has been addressed in the discussion section. (line 388-404)  

12. I recommend ensuring that the font used in each figure is uniform throughout the manuscript. Consistency in font style and size will improve the overall presentation and readability of your figures, enhancing the clarity of your findings. I also suggest enhancing the figure legend for Figure 5 to ensure it reflects a more academic tone.

 Authors’ response

This has been done for figure 1 (line 238), figure 2 (line 293), figure 3 (line 328), and figure 5 (line 729). Updated JPEG versions have also been prepared for re-submission.    

Reviewer 2 Report

Comments and Suggestions for Authors

This is an interesting study that highlights the innovative use of behavioral nudges with practical implications for supporting AMS activities in a hospital setting in a LMIC. However, I have some suggestions and comments for improving the manuscript:

-          Please move the section "1.3. Study Ethics and Approval" to the "4. Materials and Methods" section for better alignment and clarity.

-          The interviews were conducted both individually and in groups. Please provide a discussion on how this variation in the format of interviews could have influenced the results. Group dynamics might lead to different insights compared to individual interviews, potentially impacting the depth of responses or the type of feedback provided.

-          There is an imbalance in the characteristics of key informants between Round 1 and Round 2 interviews. In particular, there were no physicians or pharmacists involved in Round 2. How does this affect the results? Discuss the potential limitations or biases introduced due to the absence of these key stakeholders in the second round of interviews.

-          Please consider discussing whether Uganda has a universal health coverage (UHC) approach and how such a system could impact antimicrobial stewardship. A UHC could standardize therapy across the population, making it more feasible to implement AMS interventions such as antibiotic restrictions in the National Formulary. Additionally, UHC could help reduce the influence of commercial and social pressures, which are mentioned as factors affecting prescribing behavior.

-          The absence of a control group in the pilot study limits the strength of causal inferences. While the pre/post comparison suggests that the interventions had a positive impact, it is difficult to completely rule out other external factors that may have influenced prescribing behavior during the study period. Please discuss this limitation and its implications on the study’s findings.

-          Since the selected hospitals had prior engagement with AMS programs, the results may not fully reflect the conditions in hospitals that have had no prior exposure to AMS initiatives. Please discuss how this could affect the generalizability of the findings and what considerations might be needed for hospitals starting AMS programs from scratch.

-          The study focuses primarily on healthcare providers but does not address other significant factors contributing to antimicrobial resistance (AMR), such as patient behaviors or the broader socioeconomic context. Please discuss how these factors might influence the effectiveness of AMS interventions and the importance of addressing these aspects for a more holistic approach to tackling AMR.

Author Response

RESPONSES TO REVIEWER 2:

This is an interesting study that highlights the innovative use of behavioral nudges with practical implications for supporting AMS activities in a hospital setting in a LMIC. However, I have some suggestions and comments for improving the manuscript.

Authors’ response

Thank you for your positive feedback and for recognizing the potential of our study in supporting AMS in LMICs. We appreciate your time and feedback, and we have addressed your comments and incorporated suggestions to further strengthen the manuscript as follows:

  1. Please move section "1.3. Study Ethics and Approval" to the "4. Materials and Methods" section for better alignment and clarity.

Authors’ response

This has been done. (line 790-796)  

  1. The interviews were conducted both individually and in groups. Please provide a discussion on how this variation in the format of interviews could have influenced the results. Group dynamics might lead to different insights compared to individual interviews, potentially impacting the depth of responses or the type of feedback provided.

Authors’ response

Thank you for your suggestion. However, in our study, only individual key informant interviews were conducted, and no group interviews were held during the formative phase of the study. We have added a statement to emphasize this. (line 627-631).

  1. There is an imbalance in the characteristics of key informants between Round 1 and Round 2 interviews. In particular, there were no physicians or pharmacists involved in Round 2. How does this affect the results? Discuss the potential limitations or biases introduced due to the absence of these key stakeholders in the second round of interviews.

Authors’ response

Thank you for this insight and suggestion. We have added this as a limitation of our study. (line 547-551).

  1. Please consider discussing whether Uganda has a universal health coverage (UHC) approach and how such a system could impact antimicrobial stewardship. A UHC could standardize therapy across the population, making it more feasible to implement AMS interventions such as antibiotic restrictions in the National Formulary. Additionally, UHC could help reduce the influence of commercial and social pressures, which are mentioned as factors affecting prescribing behavior.

Authors’ response

Thank you for this insight and suggestion. We have added this in the discussion section. (line 357-364).

  1. The absence of a control group in the pilot study limits the strength of causal inferences. While the pre/post comparison suggests that the interventions had a positive impact, it is difficult to completely rule out other external factors that may have influenced prescribing behavior during the study period. Please discuss this limitation and its implications on the study’s findings.

Authors’ response

Thank you for this important suggestion. This has been added as a study limitation. (line 503-510).

  1. Since the selected hospitals had prior engagement with AMS programs, the results may not fully reflect the conditions in hospitals that have had no prior exposure to AMS initiatives. Please discuss how this could affect the generalizability of the findings and what considerations might be needed for hospitals starting AMS programs from scratch.

Authors’ response

This has been added as a study limitation. (line 516-522).

  1. The study focuses primarily on healthcare providers but does not address other significant factors contributing to antimicrobial resistance (AMR), such as patient behaviors or the broader socioeconomic context. Please discuss how these factors might influence the effectiveness of AMS interventions and the importance of addressing these aspects for a more holistic approach to tackling AMR.

Authors’ response

This has been added as a study limitation and as an important aspect to consider in future research. (line 535-542).

Round 2

Reviewer 1 Report

Comments and Suggestions for Authors

Thank you for your comprehensive and thoughtful responses to my comments. I appreciate the revisions you made, especially regarding the statistical analysis, the language used in interviews, the enhancements to the discussion, and the relocation of ethics approval. These changes significantly enhance the clarity and coherence of the manuscript. I am pleased to accept the manuscript in its current form.